# Schwann Cell Autophagy and Necrosis as Mechanisms of Cell Death by *Acanthamoeba*

**DOI:** 10.3390/pathogens9060458

**Published:** 2020-06-09

**Authors:** Ismael Castelan-Ramírez, Lizbeth Salazar-Villatoro, Bibiana Chávez-Munguía, Citlaltepetl Salinas-Lara, Carlos Sánchez-Garibay, Catalina Flores-Maldonado, Dolores Hernández-Martínez, Verónica Anaya-Martínez, María Rosa Ávila-Costa, Adolfo René Méndez-Cruz, Maritza Omaña-Molina

**Affiliations:** 1Posgrado en Ciencias Biológicas, Universidad Nacional Autónoma de Mexico (UNAM), Av. Ciudad Universitaria 3000, Coyoacán P.C. 04510, Mexico; ismaelc.40@gmail.com; 2Laboratorio de Amibas Anfizoicas, Facultad de Estudios Superiores Iztacala (FESI), Medicina, UNAM, Tlalnepantla 54090, Mexico; alol_madole@yahoo.com.mx; 3Departamento de Infectómica y Patogénesis Molecular, CINVESTAV-IPN, Ciudad de Mexico 07360, Mexico; bioagam@hotmail.com (L.S.-V.); bchavez@cinvestav.mx (B.C.-M.); 4Departamento de Neuropatología, Instituto Nacional de Neurología y Neurocirugía “Manuel Velasco Suárez”, Ciudad de Mexico 14269, Mexico; cisala69@hotmail.com (C.S.-L.); carlos.s.garibay@live.com.mx (C.S.-G.); 5Laboratorio de Histología y Patología, FESI, Medicina, UNAM, Tlalnepantla 54090, Mexico; 6Departamento de Fisiología, Biofísica y Neurociencias, CINVESTAV–IPN, Ciudad de Mexico 07360, Mexico; ceflores@fisio.cinvestav.mx; 7Centro de Investigación en Ciencias de la Salud, Facultad de Ciencias de la Salud, Universidad Anáhuac, Huixquilucan C.P. 52786, Mexico; anayamtz@yahoo.com.mx; 8Departamento de Neurociencia, FESI, UNAM, Tlalnepantla 54090, Mexico; nigraizo@unam.mx; 9Laboratorio de Inmunología, FESI, UNAM, Tlalnepantla 54090, Mexico; renemen@gmail.com

**Keywords:** *Acanthamoeba*, Schwann cell, autophagy, cell death, necrosis, cytopathic effect

## Abstract

Amoebae of the genus *Acanthamoeba* are etiological agents of granulomatous amoebic encephalitis (GAE). Recently, through an in vivo GAE model, *Acanthamoeba* trophozoites were immunolocalized in contact with the peripheral nervous system (PNS) cells—Schwann cells (SC). In this study, we analyzed in greater detail the in vitro early morphological events (1, 2, 3, and 4 h) during the interaction of *A. culbertsoni* trophozoites (ATCC 30171) with SC from *Rattus norvegicus* (ATCC CRL-2941). Samples were processed for scanning and transmission electron microscopy as well as confocal microscopy. After 1 h of interaction, amoebae were observed to be adhered to the SC cultures, emitting sucker-like structures associated with micro-phagocytic channels. In addition, evidence of necrosis was identified since edematous organelles as well as multivesicular and multilamellar bodies characteristics of autophagy were detected. At 2 h, trophozoites migrated beneath the SC culture in which necrosis and autophagy persisted. By 3 and 4 h, extensive lytic zones were observed. SC necrosis was confirmed by confocal microscopy. We reported for the first time the induction of autophagic and necrotic processes in PNS cells, associated in part with the contact-dependent pathogenic mechanisms of *A. culbertsoni* trophozoites.

## 1. Introduction

Free-living amoebae (FLA) of the genus *Acanthamoeba* are cosmopolitan protozoans commonly found in natural environments, they play a significant ecological role in controlling bacterial populations. Some species of this genus, in addition to being ecologically relevant, are also important in the medical field, due to their ability to exist as free-living organisms and as parasites, becoming a threat to the health and life of the hosts [1,2]. In the cornea, *Acanthamoeba* are etiological agents of *Acanthamoeba* keratitis (AK), a painful, vision–threatening infection occurring primarily in immunocompetent persons and contact lens users [3]. In the central nervous system (CNS), they are etiological agents of granulomatous amoebic encephalitis (GAE), which mostly occurs in people with metabolic, physiological, and immunological disorders, reporting greater than 90% mortality. GAE infection is characterized by a chronic protracted slowly progressive CNS, which also may involve the lungs and skin [2,4].

Clinical studies based in post–mortem tissues suggest that CNS invasion by *Acanthamoeba* trophozoites occurs by dissemination through blood flow from the primary site of infection. The primary *Acanthamoeba* site of infection can be the respiratory tract, were amoebae reach the olfactory neuroepithelium, leading to interactions with the blood–brain barrier and, finally, CNS invasion [2,5,6]. Moreover, through in vivo experimental models, blood flow dissemination of amoebae inoculated in the peritoneum [7] or intranasal route of healthy [4] and diabetic mice [8] has been reported.

In vitro experiments have been implemented in order to describe the cytopathic effect of *Acanthamoeba* in different targets, such as human corneal tissue [9], hamster corneal cells, Madin–Darby Canine Kidney cells (MDCK) [10,11], nervous cells such as neuroblastoma cells [12,13,14], microglial cells [15,16,17], and brain microvascular endothelial cells [6,18]. Pathogens penetrate either through the cell (transcellular route) or between the cells (paracellular route) [19]. We have reported that *Acanthamoeba* target tissue invasion occurs via paracellular route by targeting the tight junctions [9].

Recently, we described, by histological and immunohistochemical techniques, the early morphological events (24–96 h) during the invasion of *A. castellanii* and *A. culbertsoni* in a murine GAE model in healthy and diabetic mice, showing that trophozoites adhere to the respiratory and olfactory epithelium near the nasal turbinates, penetrating the tissue between the cellular junctions to subsequently invade the olfactory nerve bundles, Schwann cells (SC), and the base of the epineurium, with the absence of an inflammatory infiltrate, and without causing evident tissue damage. Subsequently, trophozoites invade the olfactory bulb and white matter in the central subcortical cortex of the brain, reaffirming the idea that contact-dependent mechanisms are relevant to amoebae of the *Acanthamoeba* genus, regardless of the site of invasion [4,8].

We now consider it necessary to understand the pathogenic mechanisms that these amoebae carry out on SC, which envelop the nerve sheaths in the peripheral nervous system (PNS), providing myelin that protects these structures and facilitates nerve impulse [20]. Damage to SC involves atrophy and destruction of the nerve, consequently losing motor skills, sensation, or both. With this purpose in mind, we carried out the in vitro coincubation of *Acanthamoeba culbertsoni* trophozoites with SC, in order to describe, through scanning (SEM) and transmission (TEM) electron microscopy as well as confocal microscopy (CM), the early events that take place during the interaction of these amoebae with SC, highlighting the importance of the contact-dependent mechanisms that cause damage to PNS cells and their possible impact on the pathophysiology of infection.

## 2. Results

### 2.1. Reactivation of A. culbertsoni Virulence

Reactivation of virulence was carried out through one intranasal passage of trophozoites in BALB/c mice; 40% of the inoculated mice died on the seventh day and the amoebae were recovered from the brain and the lungs. At day 21 post-infection, the surviving mice were sacrificed following the ethical protocol for handling laboratory animals. Amoebae were recovered only from the lungs (Table 1). Interactions with SC were carried out with amoebae recovered from the brains of the dead mice on the seventh day.

### 2.2. Schwann Cell Culture

By SEM it was observed that control SC incubated during 4 h in a proportion 3:1 of Dulbecco′s Modified Eagle′s Medium (DMEM):Bacto Casitone were confluent with a regular structure. Through TEM, typical organelles of eukaryotic cells and the formation of desmosome adherent junctions were observed (Figure 1). It is important to mention that the SC in study do not present tight junctions.

### 2.3. Analysis of the Interaction of A. culbertsoni Trophozoites with SC through SEM

By SEM, during the analysis of control SC cultures, no evidence of morphological alterations was observed. SC monolayer was confluent, and no evidence of damage, ulceration, pitting, or any other defects were detected (Figure 2A). During the interaction between *A. culbertsoni* trophozoites with SC, amoebae adhered to the monolayer from an early timepoint, emitting cytoplasmic extensions (Figure 2B) and producing cytopathic effect on the SC. After 2 h of interaction. it was even possible to observe the trophozoites cell division process (Figure 2C); lytic zones were evident, indicating a cytolytic effect of trophozoites on the SC. *A. culbertsoni* migrated beneath the SC, exposing the underlying substrate (Figure 2D). During all the interaction times analyzed, the amoebae emitted acanthopods and thin sucker-like structures in intimate contact with the SC, suggesting a phagocytic process. Similar projections have been reported by Gonzalez-Robles et al. [21] in *A. castellanii* and by Marciano-Cabral and Cabral [22] in *Naegleria fowleri.* Moreover, at 2 h it was possible to detect amoebae emitting more than one cytoplasmic extension of this type in contact with SC (Figure 2E).

At 3 h post-interaction, amoebae in contact with the SC cytoplasmatic prolongations were observed, persisting the phagocytic process (Figure 2F); trophozoites were also located in areas close to the substrate in which culture damage was evident (Figure 2G). Finally, after 4 h of interaction, extensive lytic areas were visualized (Figure 2H).

### 2.4. Analysis of the Interaction of A. culbertsoni Trophozoites with SC through TEM

Results obtained by SEM were corroborated by TEM. As expected, control cultures cells were observed in optimal conditions, highlighting the presence of desmosome adherent junctions (Figure 3A). In the experimental cultures, since the first hour post-interaction, trophozoites were observed in intimate contact with SC through acanthopodium, suggesting the formation of endocytic structures (Figure 3B–D). Trophozoites frequently were observed in contact with SC at least in two zones (Figure 3D). Moreover, in one of them, it was possible to observe a micro-phagocytic channel, visualizing the content that was directed towards the digestive vacuole (Figure 3E).

The presence of adherent junctions was frequently observed on the SC (Figure 4A). In addition, ultrastructural changes in the Golgi apparatus (Figure 4B), as well as lipofuscin granules (Figure 4C) and multilamellar and multivesicular bodies on the SC (Figure 4D), were evident.

After 2 h post-interaction, ultrastructural alterations in the rough endoplasmic reticulum (Figure 5A) and mitochondria (Figure 5B) were observed. Multilamellar bodies with the characteristic double membrane persisted (Figure 5C and Appendix A), as well as with a single membrane (Figure 5D). After 3 h post-interaction, processing of the samples by TEM was not possible, due to the extensive lytic zone caused by the amoebae to the SC cultures.

### 2.5. A. culbertsoni Induces Necrosis but Not Apoptosis in SC

To determine if *A. culbertsoni* was able to induce apoptosis or necrosis in SC, we carried out amoebae–SC interactions, which were stained with Annexin V/propidium iodide and analyzed by CM. In the early stages of apoptosis, phosphatidylserine is exposed on the outer face of the plasma membrane. Experimentally, this is recognized and stained in green by Annexin V-FITC. In contrast, in necrotic processes the plasma membrane is seriously damaged, allowing the passage of dyes, such as propidium iodide, which stains the nucleus. Cultures in Figure 6 were negative to Annexin V staining in all assays; however, red staining was positive, increasing as the interaction time with the amoebae increases. These results suggest that *A. culbertsoni* induces necrosis but not apoptosis on SC.

## 3. Discussion

Amoebae of the genus *Acanthamoeba* are ecologically and clinically relevant. Its potential to cause AK, skin infections, and mostly fatal infections in the CNS, such as GAE among susceptible human populations, has been recognized [2].

The pathogenesis of infections caused by *Acanthamoeba* is complex and not well understood. Several authors have carried out studies to describe the pathogenic mechanisms of these protozoa in different tissues and cell lines, such as MDCK cells [11], corneal tissue [10], neuroblastoma cells [12,13], microglial cells [15,16,17], and brain microvascular endothelial cells [6,18]; however, until now, no reports about the invasion mechanisms that these amoebae carry out in PNS cells, particularly in SC, have been published. Knowledge about the pathogenic mechanisms that these amoebae carry out in different cells is useful to improve the diagnostic and therapeutic methods to treat the various pathologies.

The strain in study, *A. culbertsoni* (ATCC^®^ 30171™), was the first FLA to be reported as a pathogenic amoeba for mammals [23]; for that reason we considered it important to reactivate its virulence in order to maintain the strain in optimal conditions. *A. culbertsoni* has shown a 40% virulence in BALB/c mice seven days post–inoculation, similar to that reported in *Naegleria fowleri* infections, a highly virulent amoeba [3]. In our study, the pathogenic and invasive capacity of *A. culbertsoni* was confirmed since the trophozoites migrated to the CNS in as soon as a few days, causing mice death, which was reaffirmed once the interactions were carried out.

During the interaction of amoebae with SC, by SEM it was observed that trophozoites cause mainly cytopathic damage through contact-dependent mechanisms, initiating with adhesion, and then followed by migration to below the monolayer, emission of sucker-like structures—leading to the phagocytosis process—and finally inducing cell death. This mechanism is similar to those reported in hamster and human corneal tissue [9,10], MDCK cells [11], as well as an in vivo GAE model [4], suggesting that *Acanthamoeba* perform similar pathogenic mechanisms regardless of the target tissue that they invade; indeed, we have reported that the phagocytosis process has been observed after trophozoites have migrated through the cellular junctions and invaded deeper areas of the corneal tissue or below the MDCK cell monolayer [9,10,11]. In spite of this, we also considered that amoebae carry out specific processes depending on the environment in which the amoebae are located, since during the interaction of *A. culbertsoni* with SC, trophozoites migrated and simultaneously caused damage to cells and induce the destruction of the monolayer in less than 4 h, probably because SC cultivated in vitro did not produce tight junctions that could facilitate the invasion and phagocytosis process. In vivo, it has been observed that these cells form autotypic tight junctions [24].

As part of the damage mechanisms, the emission of sucker-like structures is a common feature on amoebae surfaces, which carries out the phagocytosis process by pinching off small portions of the surface of the target cell [21], similar to that performed by *Naegleria fowleri*, where the phagocytosis process happens in a “piecemeal manner” [22]. Likewise, John et al. [25] argued that the number of emitted endocytic structures correlates with the virulence of this protozoa, which is in agreement with our results, since the majority of trophozoites projected more than one sucker-like structure in contact with target cells and its prolongations.

Moreover, by TEM it was possible to observe the formation of a micro-phagocytic channel that carries the cellular content of the target cell to the digestive vacuoles. Similar channels to those described in our work have been reported in other protozoa, such as *Entamoeba histolytica* [26], *Trichomonas vaginalis* [27], as well as *Acanthamoeba castellanii* [21]. Pettit et al. [13] reported in *A. culbertsoni* the presence of these structures associated with a food cup. In our work, the formation of micro-phagocytic channels was observed within small endocytic projections (sucker-like structures) in intimate contact with the SC, through which they ingested portions of these cells. It is important to underline that amoebae of the genus *Acanthamoeba* produce different forms of cytoplasmic extensions to phagocytize as part of the damage mechanisms to target cells [10,21]. Although, in our study, we observed that amoebae emitted only one type of endocytic structure, the sucker–like structure; it is not discarded that the amoeba under study can emit other endocytic projections since it has the ability to rearrange its cytoskeleton, which has also been reported in other species of the genus [28].

Moreover, simultaneous contact of various endocytic structures on a target cell may accelerate the cell death process. Therefore, during SC phagocytosis, trophozoites could modify the permeability of the cell’s plasma membrane causing organelles edema [29], inducing cell death by necrosis; this was confirmed by TEM, since endoplasmic reticulum and mitochondria edema were documented after in vitro interaction with *A. culbertsoni.* Based on these results we suggest that the amoebae in our study induced death through necrosis of the SC.

Similarly, during the analysis of images obtained by TEM, autophagic structures, such as multilamellar bodies or autophagosomes and lipofuscin granules, were observed on SC incubated with *A. culbertsoni*. Autophagy was probably a consequence of the persistent stress exerted during the interaction of trophozoites with these PNS cells. According to Yonekawa and Thorburn [30], autophagosomes are indicators of survival mechanisms through induction of autophagy and when this effort fails, cell death occurs. It is also considered that lipofuscin granules formation is the result of aggregates of undigested cellular materials by the autophagy process [31,32].

Apoptosis has so far been the best-described programmed cell death; however, recently autophagy has been proposed as another type of active or programmed cell death, characterized by the presence of autophagosomes or multilamellar bodies [30], which were observed in the SC in our study.

Simple autophagy or macro-autophagy is characterized by the presence of multilamellar bodies; besides, micro-autophagy lysosomes sequester portions of the cytoplasm, creating multivesicular bodies [30,33]. Multivesicular bodies are also considered a kind of late endosome that participates as an intermediary in the macro-autophagic process [34]. In our study, it was possible to observe multivesicular bodies, suggesting that regardless of whether it is micro or macro-autophagy, the SC dies as a consequence of this process.

Although Martinet et al. [35] argue that the ideal methods for monitoring autophagy in tissue do not exist, confocal microscopy and various molecular techniques have become the leading approaches to study autophagy in many different settings, even though they often fail to provide firm conclusions if not combined with TEM. Indeed, the authors highlight that electron microscopy is still “vital” to confirm and verify results obtained by other methods, in order to provide novel knowledge that would not have been obtained by any other experimental approaches [36]. It is important to implement subsequent studies to further demonstrate autophagy induction by the *Acanthamoeba* genus on SC as well as other target cells, including corneal, skin, and nervous cell lineages.

Even though it has been reported that *Acanthamoeba* is able to cause cell death through apoptotic [12,13,37,38,39] and necrosis mechanisms [13,15], until now it had not been reported that amoebae induce autophagic cell death.

Due to the TEM technique sometimes not detect the early stages of the apoptotic process [40,41], added to the fact that there was the antecedent of apoptosis induction by *Acanthamoeba*, we analyzed if the SC that interacted with trophozoites showed early apoptosis. As exposed in the methodology, the amoebae were interacted with SC at different timepoints and stained with Annexin V/propidium iodide and analyzed under a confocal microscope. According to the TEM analysis, we did not detect apoptosis in the SC; however, the time-dependent necrotic process was confirmed in control samples.

Pettit et al. [13] reported that *Acanthamoeba culbertsoni* (the same strain as in this study) was able to induce necrosis as well as apoptosis in murine neuroblastoma cells due to different cytolytic factors produced by the amoeba. In our results, probably the number of prolongations emitted by the amoeba during the phagocytic process on the SC may be relevant in the way amoebae induce cell death. We suggest that if a single prolongation is in contact with the target cell, the SC, in an attempt to survive, initiate the autophagy processes; however, if phagocytosis persists, the cells fail in their attempt to survive and die in this way. On the other hand, if several prolongations are phagocytizing the same cell, the cell membrane is compromised, leading to necrosis.

In previous studies, we have described the pathogenic mechanism of different species in the *Acanthamoeba* genus, both in in vitro [9,10] and in vivo [4,8] models. The mechanisms of invasion have been characterized mainly as contact dependent. In this process, it was confirmed that organisms with primary diseases, such as diabetes, are more susceptible to systemic invasion by *Acanthamoeba*. Even by histological and immunohistochemical techniques it was possible to observe immunolocalized trophozoites in contact with the SC surrounding the olfactory nerve bundles, without apparent histopathological changes [8]. However, even though the trophozoites were in contact with the SC, it was not possible to describe in more detail the interaction between the amoebae and PNS cells. In this work, through more precise techniques such as TEM and SEM, we can suggest that SC are a target cell for amoebae. In fact, after *Acanthamoeba* trophozoites cross the olfactory epithelium, one might think that the SC would be one of the first target cells of *Acanthamoeba*, invading, phagocytizing, and inducing cell death by necrotic and autophagic processes, to disable its protective function in the olfactory nerves bundles, invade them, and thus accelerate their migration through the olfactory bulb to the CNS and develop GAE, which would help explain damage to the motor functions of the body and could probably aggravate some symptoms of GAE, such as hemiparesis.

Through results shown in this study, it is probable that during the interaction of *Acanthamoeba culbertsoni* with SC, these PNS cells exert tropism on these amoebae, similar to what was reported for *Mycobacterium leprae*, *Trypanosoma cruzi*, and the Herpes Simplex virus [42].

## 4. Conclusions

In summary, this is the first study in which the in vitro cytopathic effect produced by *A. culbertsoni* in SC is reported. This interaction is characterized mainly by contact-dependent mechanisms, including the intimate contact and phagocytosis of the SC by trophozoites through the emission of cytoplasmic projections, such as “sucker-like structures”. During this interaction, it is suggested that the SC die by autophagic or necrotic mechanisms. These processes are important within the pathogenicity mechanisms of *A. culbertsoni*, and which may be useful to better understand the invasion of these amoebae into the CNS and other organs, in order to improve the diagnostic and therapeutic methods to fight these infections.

## 5. Materials and Methods

### 5.1. Amoeba Strain

The *Acanthamoeba culbertsoni* (ATCC^®^ 30171™) strain was grown and maintained in axenic culture in 2% Bacto Casitone medium supplemented with 10% fetal bovine serum (Biowest–S165H–500) and 1% antibiotic (Penicillin–Streptomycin). Trophozoites were incubated at 30 °C on 25 cm^2^ cell culture flasks.

### 5.2. Reactivation of A. culbertsoni Virulence

Intranasal inoculation in mice was carried out in order to reactivate amoeba virulence. First, trophozoites were harvested at the end of the logarithmic growth phase by chilling at 4 °C and concentrated by centrifugation for 5 min at 2500 rpm; the pellet was adjusted to obtain 1 × 10^6^ trophozoites in 20 µL of fresh Bacto Casitone medium (without fetal bovine serum and antibiotic). After that, five male BALB/c mice (3 weeks old) were anesthetized and inoculated into the nostrils, according to Culbertson et al. [43]. For 21 days, mice were fed ad libitum and monitored daily to observe some signs of infection. If mice did not develop infection or did not die after this time, they were sacrificed. Then, the brain, liver, lungs, and kidneys were placed on agar plates with non-nutritive enriched medium (NNE) to recover the amoebae. Finally, trophozoites recovered were axenized in Bacto Casitone medium in order to use in subsequent assays. At the same time, a group of five mice was inoculated with culture medium without amoebae as the control group. Experimental animals were manipulated in accordance with approved standard project number 174, for the reactivation of amphizoic amoebae virulence, supported by the Official Mexican Standard NOM–062–ZOO–1999, of technical specifications for the production, care, and use of laboratory animals, based on the Guide for the Care and Use of Laboratory Animals, published in the Official Journal of the Federation (Mexico) 2001. Experimental animals were kept at the FESI General Bioterium in microisolator systems, with a temperature-controlled environment, a light–dark cycle of 12:12, adequate food, and enough space for growth in optimal conditions.

### 5.3. Schwann Cells Culture

Schwann cells S16 (ATCC^®^ CRL–2941™) from sciatic nerve of *Rattus norvegicus* were grown and maintained on 25 cm^2^ cell culture flasks (Corning, Corning Incorporated, NY, USA) in DMEM (Sigma–D5648) supplemented with 10% fetal bovine serum and 1% antibiotics (Penicillin–Streptomycin). SC were incubated in a 5% CO_2_ atmosphere at 37 °C. The cell culture flasks were previously treated with Poly–L–lysine (Sigma–P9155) at least for 2 h. Previously to the final assays, initial experiments were performed to determine optimal in vitro cell conditions.

### 5.4. Interaction of A. culbertsoni Trophozoites with Schwann Cell

Interactions were carried out in a ratio 2:1 (SC: amoeba) under the same conditions in a mixture of DMEM:Bacto Casitone (serum and antibiotics free) in a 3:1 proportion. SC were washed three times for 5 min with 1x phosphate buffered saline (PBS), trypsinized for 15 min, and concentrated by centrifugation for 3 min at 1500 rpm; the pellet was adjusted to obtain 4.5 × 10^5^/500 µL fresh medium. SC were transferred to round plastic cover slips of 13 mm (previously treated with Poly–L–lysine and Colagen type IV (Sigma–C0543)) placed in 24-well styrene plates to perform the description by SEM. A similar process was carried out for TEM: the pellet was adjusted to obtain 1.25 × 10^6^/500 µL fresh medium and transferred to petri dishes of 35 mm × 10 mm (treated in the same way as the cover slips). Cultures were maintained in a 5% CO_2_ atmosphere at 37 °C, and after 48 h the cells were confluent. Then, cell cultures were incubated with *A. culbertsoni* trophozoites (harvested as previously described) at different times (1, 2, 3, and 4 h). The control culture was processed under the same conditions without amoebae at a longer interaction time of 4 h.

### 5.5. Scanning Electron Microscopy

After interaction, the medium was removed from the samples and then were fixed with 2.5% glutaraldehyde in 0.1 M cacodylate buffer at room temperature for 1 h. Samples were dehydrated with increasing concentrations of ethanol. Then the critical point was dried with liquid CO_2_ using a Samdri 780 apparatus (Tousimis Corp., Rockville, MD, USA) and coated with a thin layer (30 nm) of gold in an ion-sputtering device (JEOL, JFC–1100). Specimens were examined with a field-emission JEOL–JSM 7100F scanning electron microscope (JEOL Ltd., Tokyo, Japan).

### 5.6. Transmission Electron Microscopy

After co-incubation, the medium was removed from the samples and then was fixed with 2.5% glutaraldehyde in a 0.1 M cacodylate buffer, at room temperature, and post-fixed with 1% osmium tetroxide as well as dehydrated with increasing concentrations of ethanol. Samples were transferred to propylene oxide, later to a mixture of propylene oxide/epoxy resin—(1/1), (2/1), and (3/1)—and embedded in epoxy resins. Ultra-thin sections (60 nm) were obtained and stained with uranyl acetate and lead citrate. Finally, sections were observed in a JEOL JEM–1011 transmission electron microscope (JEOL Ltd. Tokyo, Japan).

### 5.7. Confocal Microscopy

To determine apoptosis, an Annexin V–FITC Apoptosis Detection Kit (Sigma–APOAF) was used. After co-incubation (in the same way as SEM), the samples were fixed with 4% paraformaldehyde for 1 h, at 37 °C at 0.5, 1, and 1.5 h. Briefly, kit instructions were followed and all samples were analyzed on a confocal laser scanning microscope (Leica, TCS SP8). As a positive control of apoptosis, the SC culture was treated with 0.8 mM H_2_O_2_ for 1.5 h [44]. The control culture was processed under the same conditions without amoebae at the longer interaction time.

## Figures and Tables

**Figure 1 pathogens-09-00458-f001:**
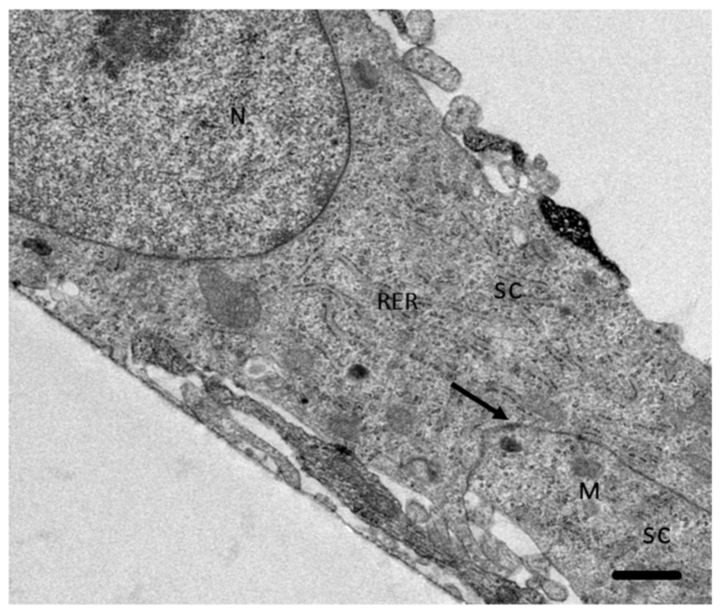
Transmission electron microscopy of the Schwann cells (SC) culture. SC have desmosome adherent junctions (arrow) as well as typical organelles of eukaryotic cells. **N:** Nucleus. **RER:** rough endoplasmic reticulum. **M:** Mitochondria. Bar = 5 µm.

**Figure 2 pathogens-09-00458-f002:**
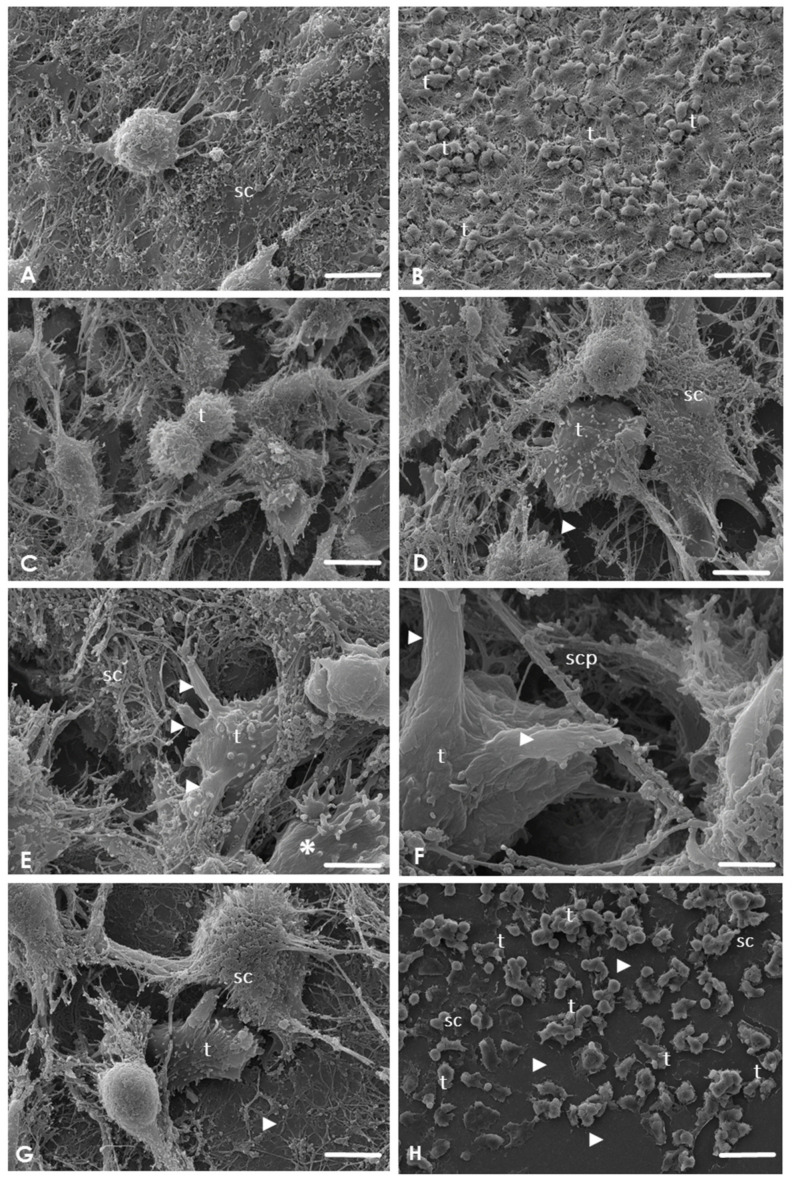
Scanning electron micrographs of the interaction of *A. culbertsoni* trophozoites with Schwann cells (SC) at different point times (1, 2, 3, and 4 h). (**A**) No morphological evidence of damage was observed in the control culture. Bar = 10 μm. (**B**) After 1 h of coincubation, trophozoites (t) were observed to be adhered to the SC surface without cell damage. Bar = 10 μm. (**C**–**E**) After 2 h of co–incubation. (**C**) Trophozoite (t) in suggestive cell division. Bar = 10 μm (**D**) Trophozoites (t) were observed migrating beneath the SC cultures, evidencing the substrate (arrowhead). Bars = 10 μm. (**E**) Trophozoite (t) in contact with SC emitting sucker–like structures (arrowheads). A trophozoite migrating (asterisk) with its typical acanthopods was also observed. Bar = 1 μm. (**F**–**G**) After 3 h of co–incubation. (**F**) Amoebae were not only in contact with the cell body but were also in contact with the cytoplasmic prolongations of SC (scp) by sucker-like structures (arrowhead). Bar = 1 μm. (**G**) Trophozoites (t) in areas close to the substrate. Culture damage was evident (arrowhead). Bar = 1 μm. (**H**) After 4 h of co-incubation. Extensive lytic zones (arrowhead) were observed in SC cultures by *Acanthamoeba* trophozoites (t). Bar = 10 μm.

**Figure 3 pathogens-09-00458-f003:**
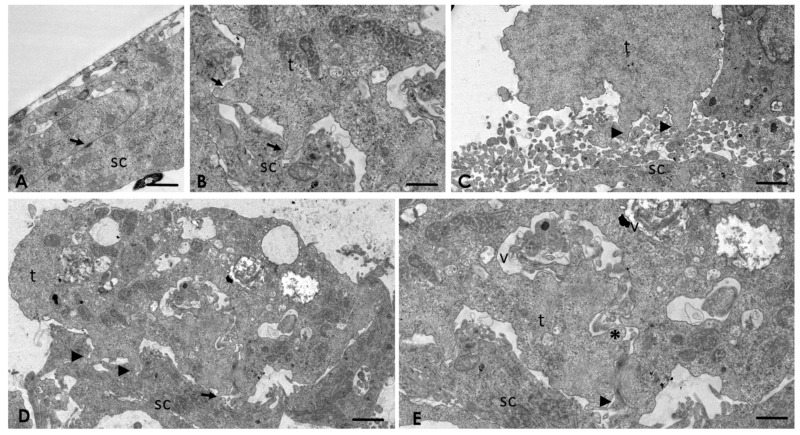
Transmission electron micrographs of the interaction of *A. culbertsoni* trophozoites with Schwann cells (SC) at 1 h. (**A**) Control. SC were maintained in culture medium without amoebae for 4 h. No morphological evidence of damage was observed. Adherents junctions (arrow), an important feature of SC cultures, were observed. Bar = 2 μm. (**B**) After 1 h of interaction, amoebae were observed in intimate contact with a SC through acanthopodium (arrows), suggesting the formation of endocytic structures. Bar = 2 μm. (**C**) The formation of probable endocytic structures (arrowhead) was a frequent phenomenon. Bar = 5 μm. (**D**) *A. culbertsoni* contacted SC in different areas (arrowhead and arrow); in addition, a trophozoite ingesting portions of a SC (arrow) was observed. Bar = 5 μm. (**E**) Higher magnification of the phagocytic process (arrowhead) in which cytoplasmic material reaches the digestive vacuoles (V) through a micro-phagocytic channel (asterisk). Bar = 2 μm.

**Figure 4 pathogens-09-00458-f004:**
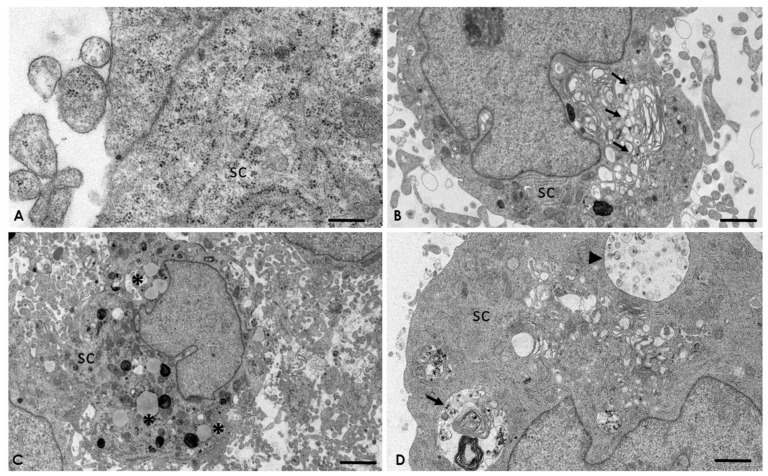
*A. culbertsoni* induces cell death on Schwann cells (SC) after 1 h of interaction. (**A**) Control. No morphological evidence of damage was observed. Bar = 1 μm. (**B**) Ultrastructural alteration of the Golgi apparatus (arrows) on SC due to edema, suggesting cell death by necrosis. Bar = 5 μm. (**C**) Lipofuscin granules (asterisks). Bar = 2 μm. (**D**) Multilamellar (arrow) and multivesicular (arrowhead) bodies, suggesting autophagy process. Bar = 2 μm.

**Figure 5 pathogens-09-00458-f005:**
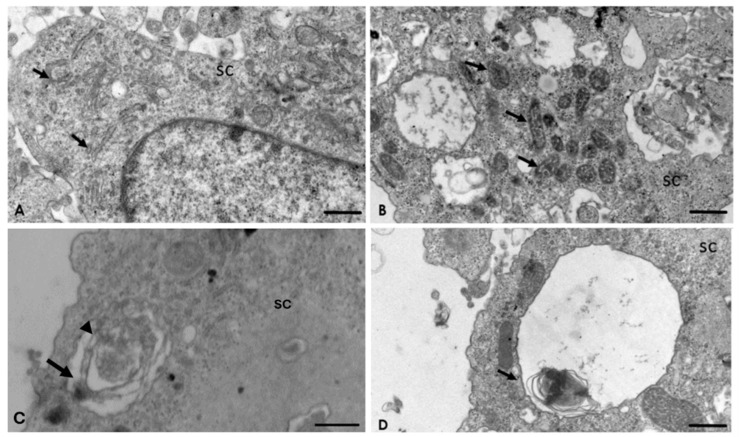
Interaction of *A. culbertsoni* and Schwann cells (SC) after 2 h of interaction. (**A**,**B**) Ultrastructural alterations in organelles of SC were frequently observed (arrows) in the rough endoplasmic reticulum (**A**) and mitochondria (**B**) Bars = 2 µm. (**C**,**D**) Multilamellar bodies with cytoplasmic material (arrows) persisted. In electron micrograph (**C**) Bar = 500 nm, a double membrane (arrow head) is evident, characteristic of these structures. Bar = 2 µm.

**Figure 6 pathogens-09-00458-f006:**
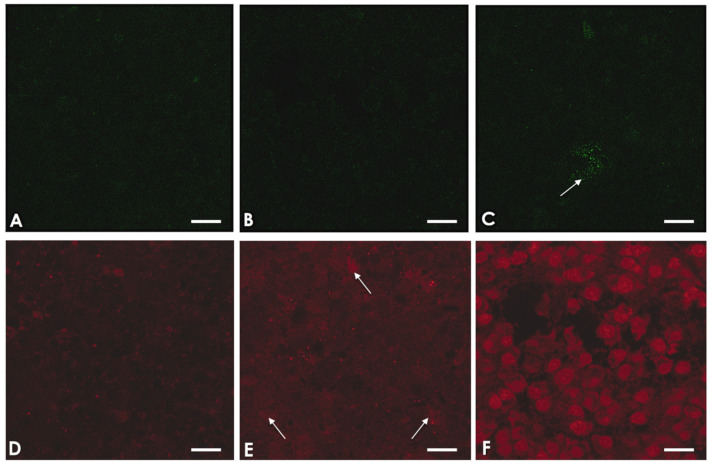
*A. culbertsoni* induces necrosis but not apoptosis on Schwann cells (SC). (**A**) Fluoresceinated Annexin V-FITC staining shows no signal on SC control cultures, (**B**) nor in those SC that interacted with *A. culbertsoni* for 1.5 h. (**C**) Staining of an apoptotic SC (arrow). (**D**) SC control do not show stained nuclei with propidium iodide. (**E**) Red-stained nuclei begin to appear (arrows) after 1 h of interaction with *A. culbertsoni* trophozoites. (**F**) After 1.5 h of interaction, abundant and clearly defined nuclei stained in red are observed. The results suggest that *A. culbertsoni* trophozoites induce necrotic processes in SC. Bars = 15 μm.

**Table 1 pathogens-09-00458-t001:** Amoebic recovery in organs.

Mice	Day of Death (d)/Sacrifice (s)	Organs	
		Brain	Lungs
1	7 (d)	+	+
2	7 (d)	+	+
3	21 (s)	−	+
4	21 (s)	−	+
5	21 (s)	−	+

(+) Recovery (−) No recovery.

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
