# Peer review of "Schwann Cell Autophagy and Necrosis as Mechanisms of Cell Death by Acanthamoeba"

_pathogens, 2020, doi:10.3390/pathogens9060458_

Round 1

Reviewer 1 Report

General comments

The study is original, valuable and the obtained results gain interest. Broad spectrum of microscopical methods were used to visualize the results which is very strong part of this manuscript. Included photos are of the highest quality and value. There are some minor changes requested in the specific comments section. In the reviewer’s opinion the presentation needs also some improvement in language.

Specific comments

  • Please update the abstract section to emphasize the importance of the studies. Please also clearly state in the abstract what was the aim of the study.

Line 31 – please update: „at early stages” of interaction?

Line 33 – “since 1 h of interaction” to be changed to “after 1h of interaction”

 Lines 34-35 – please correct as follows:  “In addition, edematous organelles characteristic for necrosis as well as multivesicular and multilamellar bodies markers of autophagy were detected”

Line 45-46  - please correct the sentence: “To human pose a threat health and life [1,2].”

Line 60 – please remove citation [11] as it is already in line 61.

  • Introduction - The aim of the study should better emphasize the novelty and originality of the performed experiments.

  • Results

2.1 – please clarify in the text if the observed interactions were carried out with the amoebae recovered from the brain after 7 or 21 days of inoculation, as this is now not clear for the reviewer.

2.2 please refer in the text to the Figure 2A.

Fig.1 if possible, please enlarge the photos to be make them more clear for the reader. They are of so great value that should be much more exposed. Reviewer also suggest to start with the control as A) – to be consistent with the Fig2.

Fig.2- Reviewer suggest to add clearer borders, maybe colored in black or wider a bit, between photos as now they merge with each other.

Fig.5 – the photos should be brightened and also scale should be added.

  • Discussion

Lines 256-262 – reviewer suggests to shift this paragraph to the separated section named “conclusions”.

Reviewer 2 Report

The manuscript presents changes in cultured Schwann cell morphology as determined by TEM and SEM and confocal microscopy following treatment with Acanthamoeba culbertsoni at 1, 2, 3, and 4 h post-treatment.  The authors suggest that A. culbertsoni can cause both necrotic and apoptotic cell death of cultured Schwann cells and that autophagy may also be involved in cell death.

Results:

Sections 2.1 and 2.2 are not results, but methodology and should not be stated in results section (ie. there are no figures for these).  You could perhaps summarize the pathogenicity of the A. culbertsoni at the beginning of what is currently 'Section 2.3.'

Section 2.3:  For this reviewer, the use of the term 'sucker' is not convincing.  No real evidence from the images that they extensions have some indentation at the tip that would make them look more like a sucker, although I am not as familiar with this feature in Acanthamoeba. It does not seem to match the image presented in reference #24. 

As such, Figure 1F is not any more convincing than Figure 1E.  Why not included a zoomed out view at 4 h (not shown) to convince us of the areas of "extensive areas of necrosis" (caption for 1G).  As such, that is not convincing either.

Figure 2D.  Why is this panel necessary when 2E illustrates this a little better?  Panels could be configured better without it.  What is a "phagocytic prolongation structure"? (line 119)  

Figure 5 - are there positive controls for apoptosis vs. necrosis? The selected cells in panel D do not seem to have any stronger of a signal than the other cells.  Do they need to be magnified more?

Suggest future studies include confocal microscope for indicators of autophagy markers before any conclusions about autophagy are made.

Overall the manuscript suffers from poor English language vocabulary use and grammar.  The errors are too numerous to list and are throughout the entire document.

Should not be a space between the value for temperature and the 'degree' symbol.

Recent review on Acanthamoeba and its role in GAE published online in 2019 by Kalra SK, Sharma P, Shyam K, Tejan N, Ghoshal U.in Exp. Parasitol.

Ethical concern:  No specific animal approval (IACUC) for protocol?  Only state that it followed Official Mexican Standard.  Not sure that this is acceptable. (Were 5 mice necessary as a control or was this also part of another protocol?) 

The results may be of interest to some investigators in the field although they do not add much significance in terms of pathogenesis in that most cell lines are damaged by Acanthamoeba by necrosis and apoptosis combinations.  More evidence is needed for autophagy but may be of interest to other investigators as morphological changes to be observed in their own cell lines or with clinical Acanthamoeba isolates.

Poor English/grammar made it difficult to concentrate on the research presented.  Too many errors to enumerate/correct without re-writing.  Requires significant revision to be understandable and for me to feel comfortable recommending.

7 of 42 refs are self-citations and the majority of them appear in a 'laundry list' of cells types (ll 58-61).  Not certain that the non-nervous system cells are necessary.

Only 2 refs from 2018-2019.  Missing a recent review on Acanthamoeba GAE.

Reviewer 3 Report

Interesting and well-described study that improves knowledge on the cytopathic activity of the genus Acanthamoeba.

Round 2

Reviewer 2 Report

The authors have made significant improvements to the Results section, especially improved labeling of figures and text to accompany/support the figures. However, in Figure 6F, it is still unclear as to what the arrows are indicating, as almost all of the nuclei appear equally red.

Although improvements have been made, English language and grammar still require correction.

LL27-29 - implication that the in vivo work is the present study when it refers to a previous study
LL33-34 - poor sentence structure
L37 - 'being' not appropriate in this sentence
L38 - 'Trough' - 'Through' is misspelled (through is used 5 times in the abstract)
L47 - 'since can live'
L50 - 'In central nervous system (CNS) are etiological agents...'
Authors have added the suggested review article, yet still self-cite even though their citation (and another) are cited within the review: L53 [2,4,5,6] - references 2 and 4 are cited in review 6 with 4 being a self-citation.

Author Response

POINT 1: The authors have made significant improvements to the Results section, especially improved labeling of figures and text to accompany/support the figures. However, in Figure 6F, it is still unclear as to what the arrows are indicating, as almost all of the nuclei appear equally red.

RESPONSE 1: We appreciate your observation and we agree with the comment in relation to figure 6, considering that the presence of the staining shown in panel 6F is homogeneous, making evident positive nuclei to propidium iodide, which implies that the SC die due necrosis. Arrows pointing to some of the nuclei could cause confusion; therefore, we decided to remove them. Highlight the reaction in full panel.

POINT 2: Although improvements have been made, English language and grammar still require correction.

RESPONSE 2: We apologize again for the errors in the writing of the manuscript, since it is not our native language. We appreciate your comment. We are sure that the revision of our manuscript by people who are fluent in English will substantially improve its quality. We inform you that we contacted the editor to request the style and grammar editing service that the Journal offers and we were informed that if the manuscript is accepted by the reviewers for publication, it will be submitted to the professional editing service of English offered by the publisher, to improve their grammar.

POINT 3: LL27-29 - implication that the in vivo work is the present study when it refers to a previous study

RESPONSE 3: The reference that we mentioned in our work describing Acanthamoeba invasion of the CNS through the GAE in vivo model, is the most important direct antecedent of our work, since we immunolocalize these amoebae in contact with Schwann cells, during their invasion route to CNS. Therefore, we consider it necessary to carry out this study and describe in more detail the interaction between Acanthamoeba culbertsoni trophozoites and Schwann cells through in vitro studies.

POINT 4: LL33-34 - poor sentence structure

RESPONSE 4: We appreciate the observation. The sentence “Through SEM, amoebae were observed adhered to SC cultures emitting sucker – like structures after 1 h of interaction, through TEM a micro – phagocytic channel was observed” was modified as follows: “After 1 h of interaction, amoebae were observed adhered to SC cultures emitting sucker-like structures associated with micro-phagocytic channels ”

POINT 5: L37 - 'being' not appropriate in this sentence

RESPONSE 5: We appreciate your suggestion. “At 2 h trophozoites migrated beneath culture cells, being more evident necrosis and autophagy” was modified as follows: “At 2 h trophozoites migrated beneath SC culture in which necrosis and autophagy persisted”

POINT 6: L38 - 'Trough' - 'Through' is misspelled (through is used 5 times in the abstract)

RESPONSE 6: We replace the word thought with some synonyms and consider that we improve the writing of the abstract

POINT 7: L47 - 'since can live'

RESPONSE 7: The sentence “since can live both as free–living organisms and as parasites” was modified as follows: “because these amoebae have the ability to exist as free–living organisms and as parasites”

POINT 8: L50 - 'In central nervous system (CNS) are etiological agents...'

RESPONSE 8: We regret the omission of the letter “s”, it was modified as follows "In central nervous system (CNS) are etiological agents of granulomatous amoebic encephalitis (GAE)”

POINT 9: Authors have added the suggested review article, yet still self-cite even though their citation (and another) are cited within the review: L53 [2,4,5,6] - references 2 and 4 are cited in review 6 with 4 being a self-citation.

RESPONSE 9: We appreciate your comments. References 2 (Kot et al., 2018) and 4 (Marciano Cabral and Cabral, 2003) have been deleted, remaining the most recent revision in which they are cited (previously 6, now 2) (Kalra et al., 2019).

It is important to mention that we have referred to previous studies in which we described the mechanisms of invasion of amoebae of the genus Acanthamoeba; which are mainly contact dependent, we initially describe the hamster and human corneal invasion where amoebae cause chronic infection, we continue with the description of early CNS invasion events and now we have implemented in the skin (in process of publication). We were surprised to note that contact-dependent mechanisms are relevant during the invasion because it was argued that the production of proteases was vital for amoebae to destroy target tissues, which we did not observe in our studies, since amoebae adhere, migrate towards cell junctions and by mechanical and or enzymatic effect, they cross into deeper layers, phagocytizing newly detached cells.

With these studies we validate processes of amoebic invasion, we are sure that if more studies were reporting similar processes then we would gladly include them in this manuscript, but this is not the case.

Our self-cites are intended to support invasion mechanisms independent of the target tissue and although they have particularities, they now allow us to suggest processes carried out by the Acanthamoeba genus and not isolated species.

Round 3

Reviewer 2 Report

As long as the authors proceed with the English editing service provided by the journal, the manuscript may be acceptable.

The manuscript still requires extensive editing.  In lines 27-91 alone there were 16 errors/edits required (typos, missing words, incorrect sentence structure, run-on sentences).  In some cases, the wording and sentence structure makes the meaning of the sentence unclear to the reader.  Example, L80ff "Its alteration leads to atrophy and destruction...."  It is unclear as to whether "Its" is referring to Schwann cells or nerve sheaths.  As I read the manuscript, there are other examples of this as well.

Author Response

POINT 1: As long as the authors proceed with the English editing service provided by the journal, the manuscript may be acceptable.

The manuscript still requires extensive editing. In lines 27-91 alone there were 16 errors/edits required (typos, missing words, incorrect sentence structure, run-on sentences). In some cases, the wording and sentence structure makes the meaning of the sentence unclear to the reader.  Example, L80ff "Its alteration leads to atrophy and destruction...."  It is unclear as to whether "Its" is referring to Schwann cells or nerve sheaths.  As I read the manuscript, there are other examples of this as well.

RESPONSE 1: We appreciate the time spent reviewing this manuscript, your suggestions for change have surely improved it substantially. We inform you that the academic editor has been contacted so that the manuscript can be submitted to the professional service of English language edition offered by the journal, in case of being accepted.

The phrase “Its alteration leads to atrophy and destruction of the nerve with loss of motor, sensory, or both” was modified as follows: “Damage to SC involves atrophy and destruction of the nerve, consequently losing motor skills, sensation, or both.”